# SINAI: Strategic Injection of Noise for Adversarial Defense with Improved Accuracy–Robustness Tradeoffs

## Abstract

Vision models are widely used for edge deployment, but they are highly vulnerable to query-based black-box adversarial attacks. Existing noise injection defenses, while promising, often overlook the unique characteristics of vision models, such as the heterogeneity between attention and feedforward modules, which fundamentally shape how injected noise propagates. Thus, previous methods yield suboptimal trade-offs between clean and robust accuracy. In this work, we scrutinize noise injection methods to vision models and provide two insights: (1) noise injection should explicitly consider the activation function, as attention modules with Softmax respond differently from FFN and CNN modules with GeLU or ReLU, and (2) gradient norms vary across logits within the same layer, so uniform noise injection for all the logits can perturb high-gradient logits and hurt clean accuracy, which motivates a fine-grained, logit-aware allocation strategy. Building on these observations, we propose an adaptive noise injection defense that combines module-level and logit-level noise allocation: injecting stronger noise to attention with softmax, while applying fine-grained, logit-aware noise in GeLU/ReLU based FFN and convolutional modules. We then formulate the search process of noise injection hyperparameters as a constrained optimization problem, in which the clean accuracy drop is bounded, and solve it via Bayesian optimization. Experiments on ViT-B-16-224 with ImageNet show that our method improves average robust accuracy by 4.2% over feature noise defense under three 10k steps query-based black-box attacks with $\approx$2% clean accuracy drop.

## 1 Introduction

Vision models such as Vision Transformers (ViTs) and convolutional neural networks (CNNs) remain indispensable for edge deployment (Ahmed et al., 2025; Pan et al., 2022; Shu et al., 2024). Yet in such scenarios, robustness against adversarial threats is particularly critical, as they are both highly vulnerable to adversarial examples: imperceptible perturbations to input images can significantly degrade classification accuracy (Bai et al., 2021; Bhojanapalli et al., 2021; Fu et al., 2022; Mahmood et al., 2021; Mao et al., 2022). While adversarial examples are often studied under the white-box setting with full model access (Carlini & Wagner, 2017; Kurakin et al., 2016; Madry et al., 2017), this assumption is unrealistic in practice (Ilyas et al., 2018a; Guo et al., 2019), such as machine learning as a service (MLaaS) scenarios (Ribeiro et al., 2015) where models are accessible only through queries. This motivates query-based black-box attacks, which are both practical and prevalent in real-world applications (Ilyas et al., 2018a;b; Andriushchenko et al., 2020; Guo et al., 2019; Al-Dujaili & O'Reilly, 2020; Liu et al., 2019; Chen & Gu, 2020; Rahmati et al., 2020). Consequently, developing effective defenses against query-based black-box attacks is important to the reliable implementation of vision models.

In response, noise injection during inference (Qin et al., 2021; Byun et al., 2022; Hung-Quang et al., 2024) has emerged as a simple yet effective strategy. This defense method can be broadly categorized into two categories, i.e., *input noise injection* and *feature noise injection*. The first category adds Gaussian noise to each input to perturb the model's output, thereby hindering gradient estimation or random search and reducing the attack success rate of adversarial queries. The effectiveness of input noise injection is closely related to the variance of the injected noise, which directly controls

its magnitude (Qin et al., 2021; Byun et al., 2022). Beyond input-level noise, Feature Noise Defense (FND) (Hung-Quang et al., 2024) extends this idea by injecting noise into intermediate features, leveraging gradient norms to analyze the trade-off between robustness and clean accuracy, thereby achieving stronger robustness improvements.

While prior work has advanced this field, these methods exhibit several limitations, which make them suboptimal in terms of clean and robust accuracy trade-off (Tsipras et al., 2018). First, existing methods mainly target CNNs and apply uniform noise injection across the entire model, without considering the structural differences of ViTs. As a result, different modules (e.g., attention and feed-forward network (FFN)) either inject insufficient noise that fails to improve robustness, or inject too strong noise that harms clean accuracy. Our empirical evaluation in Section 3.1 shows that attention, FFN, and CNN modules exhibit different levels of noise sensitivity. Inspired by (Liu et al., 2020b), we attribute this to the distinct activation functions used in these modules, i.e., Softmax and ReLU/GeLU. This finding suggests that applying the same level of noise uniformly across modules is less effective, and that noise allocation should instead be module-aware.

Second, prior work lacks a fine-grained understanding of the clean–robust trade-off. Even for CNNs, previous studies have not thoroughly examined how noise affects the balance between clean and robust accuracy. Input-level noise injection (Qin et al., 2021; Byun et al., 2022) perturbs the entire propagation process, while feature-level noise injection introduces gradient norms to analyze layer sensitivity (Hung-Quang et al., 2024). However, these approaches remain coarse-grained, since they treat all neurons in a layer in the same way. By contrast, we define a fine-grained perspective as examining noise effects at the level of individual logits within a layer. Our empirical evaluation in Section 3.2 shows that some logits are highly sensitive while others are relatively stable, which highlights that uniform noise within a layer is inadequate and motivating logit-aware noise allocation.

To this end, we propose SINAI: Strategic Injection of Noise for Adversarial defense with Improved accuracy–robustness tradeoffs, a systematic study of how ViTs respond to noise, which enables a more effective inference-time defense strategy. First, we analyze the sensitivity of different activation functions and show that attention modules with Softmax activations can tolerate larger perturbations, whereas FFN and CNN modules with GeLU or ReLU activations degrade rapidly as noise increases, motivating module-specific noise magnitudes. Second, by probing gradient norms at the logit level, we demonstrate that uniform noise allocation within FFN/CNN layers is suboptimal, and propose a non-uniform scheme that adapts to local sensitivity for a better balance between clean and robust accuracy. Finally, rather than relying on manual tuning as in prior work (Qin et al., 2021; Byun et al., 2022; Hung-Quang et al., 2024), we reformulate noise injection as a constrained optimization problem and employ Bayesian search (Akiba et al., 2019) to automatically identify optimal configurations.

## 2 RELATED WORK

### 2.1 QUERY-BASED BLACK-BOX ATTACK

Query-based attacks assume the adversary can interact with the target model through queries while having no access to parameters or gradients, and are generally divided into score-based and decision-based methods. Score-based attacks leverage the confidence scores or probabilities returned by the model to estimate gradients. Early works such as ZOO (Chen et al., 2017) and NES (Ilyas et al., 2018a) rely on zero-order optimization, while Bandit-based attacks (Ilyas et al., 2018b) improve query efficiency by exploiting gradient priors. Later methods such as SignHunter (Al-Dujaili & O'Reilly, 2020) and Square Attack (Andriushchenko et al., 2020) further reduce query cost and strengthen attack success. Decision-based attacks instead assume access only to the final predicted label. Boundary Attack (Brendel et al., 2017) starts from a large perturbation and gradually shrinks it, and HSJA (Chen et al., 2020a) improves query efficiency via binary decision feedback. RayS (Chen & Gu, 2020) proposes a ray-searching method for hard-label attacks, and recent work introduces random sign flip strategies to further boost decision-based black-box attacks (Chen et al., 2020b). These methods demonstrate that even with limited access, query-based black-box attacks can reliably fool deep models, making them a critical threat model for evaluating defenses.

## 2.2 DEFENSE WITH NOISE INJECTION

Noise injection has been widely studied as a robustness mechanism (Liu et al., 2018; He et al., 2019; Pinot et al., 2019; Xiao et al., 2020; Wu et al., 2020; Jeddi et al., 2020; Lecuyer et al., 2019; Xie et al., 2017; Dhillon et al., 2018). Early works primarily focused on white-box attacks and typically relied on additional adversarial training. However, subsequent studies revealed that many of these defenses offered only a false sense of robustness, stemming from gradient obfuscation, which can be easily circumvented by gradient estimation techniques under white-box settings. (Athalye et al., 2018). More recent efforts have shifted toward training-free defenses in black-box settings: Random Noise Defense (RND) (Qin et al., 2021) and Small Noise Defense (SND) (Byun et al., 2022) inject Gaussian noise to the input at inference time, while Feature Noise Defense (FND) (Hung-Quang et al., 2024) extends this idea by injecting noise into intermediate features and analyzing the effect through gradient norms to better understand the trade-off between robustness and clean accuracy. Although these approaches provide both theoretical justification and empirical robustness against query-based attacks, they generally overlook the better clean–robust trade-off. In contrast, our work identifies the limitations of uniform and coarse-grained noise injection and introduces strategic noise injection that leverages module- and logit-aware differences to achieve a better balance.

## 3 MOTIVATION

In this section, we introduce the motivation behind our method. In Section 3.1, we empirically demonstrate that different modules exhibit heterogeneous sensitivity to injected noise. Subsequently, in Section 3.2, we highlight the necessity of adopting a more fine-grained noise for individual logits.

### 3.1 MODULE-LEVEL HETEROGENEITY IN NOISE SENSITIVITY

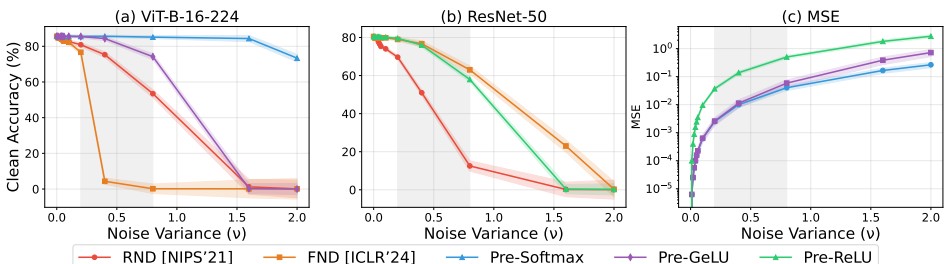

Figure 1: Grey shadow is the obvious region of noise injection. (a) ViT-B-16-224 evaluation under various noise variances with noise injection to different activation functions. (b) ResNet-50 results under various noise variances. (c) Mid-layer MSE of different activation functions under noise, showing different sensitivity of Softmax, GeLU, ReLU.

As discussed earlier, existing noise-based defenses primarily target CNNs and adopt a uniform injection strategy across the entire model, without considering the structural heterogeneity of ViTs (Qin et al., 2021; Hung-Quang et al., 2024). To investigate whether different modules in ViTs exhibit distinct levels of noise resilience, we draw inspiration from prior findings that different activation functions have varying noise sensitivities (Liu et al., 2020b). We therefore conduct an empirical study by injecting noise into each module individually based on its activation function. Specifically, we evaluate ViT-B-16-224 (Dosovitskiy et al., 2020)(with Softmax in the attention layers and GeLU in the FFN), as well as ReLU in ResNet-50 (He et al., 2016), using the ImageNet dataset (Russakovsky et al., 2015).

Unlike prior work (Hung-Quang et al., 2024) that injects noise at the layer's output feature, we introduce Gaussian noise directly into the *pre-activation logits* within each module. Formally, given a linear transformation $z = Wx + b$, we inject noise to it as $\tilde{z} = z + \varepsilon$ with $\varepsilon \sim \mathcal{N}(0, \sigma^2)$ and then apply the nonlinearity $a = f(\tilde{z})$. This noise injection ensures that the perturbation interacts with the activation function, thereby exposing heterogeneous behaviors across modules. Figure 1 presents the clean accuracy under such injection, and we further quantify the internal responses by computing the mean squared error (MSE) of each module at the mid-layer.

**Insights.** The results clearly reveal different noise sensitivity across modules. For ViT-B-16-224 (Figure 1(a)), injecting noise before the Softmax in attention leads to only a minor accuracy drop even under large variance. The MSE trend in Figure 1(c) further confirms its stability, as errors accumulate much more slowly than in other modules. In contrast, nonlinear activations such as GeLU in the ViT FFN and ReLU in ResNet-50 are more sensitive. Clean accuracy degrades sharply even at small noise variance, and their MSE grows orders of magnitude faster, reflecting the strong interaction between noise and nonlinear gating. These observations suggest that a uniform noise injection strategy is fundamentally suboptimal: modules differ in their noise sensitivity. This motivates noise allocation across modules, which we will theoretically justify in Section 4.2.1.

## 3.2 GRADIENT-NORM ANALYSIS OF INDIVIDUAL LOGIT

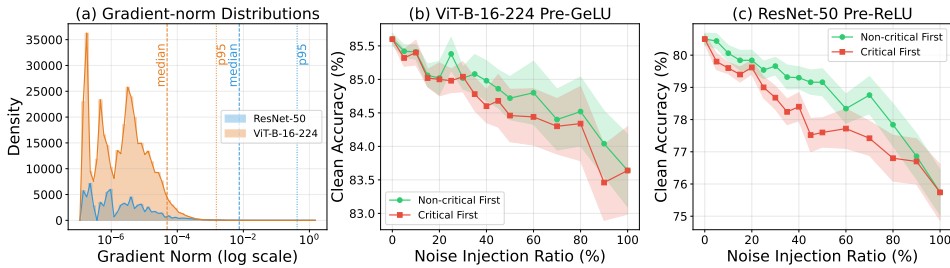

Figure 2: (a) Gradient-norm distributions before GeLU (ViT-B-16-224) and ReLU (ResNet-50), showing median and 95th percentile. (b) ViT-B-16-224 Pre-GeLU and (c) ResNet-50 Pre-ReLU: clean accuracy under noise injection, comparing non-critical-first vs. critical-first strategies.

Theoretical analysis (Hung-Quang et al., 2024) indicates that the clean accuracy of a randomized model depends on the accumulated gradient norm of the selected layer and the variance of the injected noise. Building on this insight, we empirically examine the clean accuracy degradation of GeLU and ReLU activation functions to assess whether layer-level gradient norms alone are sufficient to guide effective noise injection.

Figure 2(a) shows the gradient-norm distributions across mid-layers of ResNet-50 and ViT-B-16-224: ViT-B-16-224 generally exhibits lower gradient norms, whereas ResNet contains a heavier tail, indicating fewer logits with large gradients. Prior work on activation sparsity (Li et al., 2022; Kurtz et al., 2020; Song et al., 2024) has shown that not all activations contribute equally to predictions; in fact, important activation patterns with large gradient norms vary dynamically across inputs (Liu et al., 2020a; Li et al., 2022; Liu et al., 2021) and form critical pathways for accurate predictions (Yu et al., 2018; Khakzar et al., 2021). Motivated by this, we compare the effect of injecting noise into critical pathways (i.e., logits with large gradient norms) versus non-critical pathways (i.e., logits with small gradient norms). Our results in Figure 2(b) and (c) demonstrate that injecting noise into non-critical logits leads to substantially smaller drops in clean accuracy compared to injecting into critical logits, underscoring the need for fine-grained noise injection strategies.

**Insights.** These results highlight three important points: (1) the heterogeneity across logits makes it necessary to consider gradient norm at the individual-logit level, and (2) noise allocation that prioritizes non-critical logits yields a better clean accuracy. (3) ResNet-50's heave-tailed gradient distribution amplifies the gap between non-critical first and critical first noise injection. In the following section, we will provide a formal explanation of why gradient norm governs the clean accuracy and robustness, and why our selective strategy improves clean-robust trade-offs.

## 4 METHOD

In this section, we first introduce our noise injection strategy in Section 4.1, motivated by our earlier observations, then present the theoretical support in Section 4.2, and finally provide the optimization framework for searching the best noise injection configuration in Section 4.3.

### 4.1 STRATEGICAL NOISE INJECTION

We design noise injection strategies that adapt to both module-level and logit-level characteristics.

**Module-level: different noise magnitudes.** Our analysis in Section 3.1 shows that different modules exhibit distinct noise sensitivities: attention modules with Softmax activations are more noise-tolerant, while FFN and CNN modules with GeLU or ReLU activation functions are more fragile. Motivated by this, we inject noise with larger variance into attention modules and weaker noise into FFN/CNN modules, rather than applying a uniform magnitude across the entire model.

**Logit-level: selective noise injection.** Our analysis in Section 3.2 shows that gradient norms vary across logits within the same layer, and accumulating gradient norm from non-critical logits achieves a subtle clean accuracy degradation. This indicates that uniform noise allocation within a layer is suboptimal, motivating a selective strategy that prioritizes injecting noise into non-critical logits. Formally, the gradient of loss $L$ with respect to a pre-activation $z_i$ can be written as $\frac{\partial L}{\partial z_i} = \frac{\partial L}{\partial a_i} \phi'(z_i)$, where $a_i = \phi(z_i)$ and $\phi(\cdot)$ is the nonlinearity. The gradient norm $\|\nabla_z L\|$ determines the sensitivity of the loss to perturbations at this unit. Note that $\frac{\partial L}{\partial z_i}$ depends on both the upstream gradient $\frac{\partial L}{\partial a_i}$ and the local Jacobian $\phi'(z_i)$. Following the analysis in previous work (Hung-Quang et al., 2024), we make a *statistical isotropy assumption*: after normalization layers such as Batch Normalization/Layer Normalization, the upstream gradients across different dimensions can be treated as approximately i.i.d. with equal variance, and are weakly correlated with the logit $z_i$. Under this assumption, the variation of the gradient norm across units is dominated by the Jacobian factor $\phi'(z_i)$, which directly links the logit value $z_i$ to the likelihood of having a large gradient norm.

For ReLU, $\phi'(z) = \mathbf{1}_{z>0}$, so negative logits are guaranteed to produce zero gradients and thus can be safely noised. For GeLU, $\phi'(z)$ is a monotone increasing function of $z$, implying that larger logits are statistically more likely to have larger gradient norms. Therefore, ranking logits by their values provides a simple yet effective proxy for ranking by gradient norm.

Based on this proxy, we design a selective noise injection strategy: (i) identify critical logits with large gradient norm and avoid injecting noise into them, and (ii) inject noise to non-critical logits with small gradient norm. This non-uniform allocation achieves a better clean–robust trade-off compared to uniform noise injection.

## 4.2 THEORY JUSTIFICATIONS

We provide theoretical foundations for our method by first analyzing the noise sensitivity of different activation functions (Section 4.2.1), and then modeling the clean–robust trade-off under noise injection and proving that injecting into small-gradient logits is better (Section 4.2.2).

### 4.2.1 ACTIVATION FUNCTION'S SENSITIVITY TO NOISE

First, we introduce the definition of noise sensitivity for activation functions. We inject Gaussian noise before the activation, $\tilde{z} = z + \varepsilon$ with $\varepsilon \sim \mathcal{N}(0, \sigma^2)$. For an activation function $a(\cdot)$, its sensitivity to noise can be captured by the first terms of the Taylor expansion: $a(z + \varepsilon) \approx a(z) + a'(z)\varepsilon + \frac{1}{2}a''(z)\varepsilon^2$. Taking expectation with $\varepsilon \sim \mathcal{N}(0, \sigma^2)$, we have $\mathbb{E}[\varepsilon] = 0$ and $\mathbb{E}[\varepsilon^2] = \sigma^2$. Thus the first-order term $a'(z)\varepsilon$ vanishes in expectation, leading to

$$\mathbb{E}[a(\tilde{z})] \approx a(z) + \tfrac{1}{2}\sigma^2 a''(z), \qquad \mathrm{Var}[a(\tilde{z})] \approx \sigma^2 (a'(z))^2.$$

Here, $a'(z)$ determines how much the variance is amplified, while $a''(z)$ determines how much the mean is shifted. An activation function is considered more noise-sensitive when either one is large. Based on this definition, we next analyze the behavior of specific activation functions.

**Softmax.** For Softmax, $s_i(\mathbf{z}) = \exp(z_i)/\sum_j \exp(z_j)$, sensitivity is characterized by the Jacobian

$$J_{ij} = s_i(\delta_{ij} - s_j).$$

The Jacobian norm is bounded (e.g., $\|J\|_2 \leq 1/2$ in the binary case) and decreases as the logit margin grows, so variance from noise is strongly suppressed when one class is dominant. Moreover, Softmax is translation-invariant, i.e., $\mathrm{softmax}(\mathbf{z}+c\mathbf{1}) = \mathrm{softmax}(\mathbf{z})$, which cancels common-mode noise. Hence Softmax exhibits the slowest growth in output perturbation as $\sigma^2$ increases, making it the most noise-resilient among the activations we study.

**ReLU.** For ReLU, $a(z) = \max(0, z)$, the function is non-differentiable at $z = 0$, so the Taylor approximation does not apply exactly. When $z > 0$, noise passes through directly, i.e., $\mathrm{Var}[a(\tilde{z})] \approx \sigma^2$, while for $z < 0$ the output is clamped near zero and noise is suppressed. The dominant effect arises at the switching boundary $z \approx 0$, where noise can flip the activation. Formally, a flip occurs when $\tilde{z} = z + \varepsilon < 0$, i.e., $\varepsilon < -z$ with $\varepsilon \sim \mathcal{N}(0, \sigma^2)$. Standardizing $\varepsilon$ yields

$$p_{\mathrm{flip}} = \Pr[\varepsilon < -z] = \Phi(-z/\sigma),$$

where $\Phi(\cdot)$ is the standard Gaussian cumulative distribution function. As $\sigma^2$ increases, this probability rises, producing abrupt output changes and making ReLU the most sensitive to noise.

**GeLU.** For GeLU, $a(z) = z\Phi(z)$, the function is smooth and differentiable. Its derivative is $a'(z) = \Phi(z) + z\phi(z)$, which lies mostly in $(0, 1)$. The curvature is $a''(z) = \phi(z)(2 - z^2)$, which is nonzero around $z \approx 0$. Thus, noise is partially reduced compared to ReLU due to bounded slope, but the nonzero curvature introduces a mean shift near the zero point. This makes GeLU moderately sensitive to noise.

In summary, as noise variance increases, output perturbations grow fastest for ReLU, at a moderate rate for GeLU, and slowest for Softmax, consistent with our findings in Section 3.1.

### 4.2.2 TRADE-OFF MODELING AND NOISE ALLOCATION FOR BETTER TRADE-OFF

Recent work (Hung-Quang et al., 2024) demonstrates that injecting random noise into hidden features enhances robustness against query-based attacks, as it disrupts the attacker's gradient estimation or search direction. The likelihood of incorrect updates from attackers increases with both the magnitude of the injected noise and the gradient norm at the injection point.

**Robustness improvement.** Following Theorem 1 in FND (Hung-Quang et al., 2024), the attacker's wrong-update probability is a *monotone increasing* function of $\frac{2}{\mu} \cdot \frac{\sum_j \sigma_j^2 (\partial_{z_j}(L \circ g))^2}{\|\nabla_x (L \circ f)\|_2^2}$, where $\mu$ is the probing scale. Absorbing constants into per-logit coefficients, we define $b_j \triangleq \mathbb{E}\left[(\partial_{z_j}(L \circ g))^2\right]$, and summarize robustness as

$$\mathcal{R}(\sigma^2) = \sum_{j=1}^m b_j \sigma_j^2, \tag{1}$$

which grows monotonically with the weighted sum of per-logit variances, where $b_j$ reflects the squared gradient component at logit $j$.

**Clean accuracy degradation.** Clean accuracy depends on the margin rescaled by the standard deviation of the linearized perturbation (Hung-Quang et al., 2024), with per-logit covariance, we have

$$\mathrm{Acc}_{\mathrm{clean}} \propto \frac{L(f(x), y)}{\sqrt{\sum_{j=1}^m \sigma_j^2 (\partial_{z_j}(L \circ g))^2}}. \tag{2}$$

For small noise, a Taylor expansion yields the linear approximation

$$D_{\mathrm{clean}}(\sigma^2) \approx \sum_{j=1}^m a_j \sigma_j^2, \tag{3}$$

where $a_j$ scales with $(\partial_{z_j}(L \circ g))^2$; the detailed expression is given in Appendix A.

**Optimization formulation.** Combining Equation (1)–(3), we allocate per-logit variances via

$$\max_{\sigma_j^2 \geq 0} \sum_{j=1}^m b_j \sigma_j^2 \quad \text{s.t.} \quad \sum_{j=1}^m a_j \sigma_j^2 \leq C_0, \tag{4}$$

where $a_j, b_j$ are gradient-dependent (both $\propto (\partial_{z_j}(L \circ g))^2$ up to a margin factor).

**Assumption.** For a fixed model/attack family, per-logit robustness coefficients vary slowly: $b_j = \bar{b}(1 + \epsilon_j)$ with $|\epsilon_j| \leq \epsilon \ll 1$.

**Theorem 4.1** (Small gradient first). *For Equation (4), the KKT conditions select logits by the ratio $\rho_j = b_j/a_j$. If $b_j$ varies little across logits, this is equivalent to ranking by $a_j$. Since $a_j \propto (\partial_{z_j}(L \circ g))^2$, the optimal allocation is to inject noise into logits with the smallest gradients; the rule is exact when $b_j$ is constant and $(1 - O(\epsilon))$-optimal otherwise.*

*Proof sketch.* KKT gives $\sigma_j^2 > 0 \Rightarrow b_j = \lambda a_j$, i.e. select by $\rho_j = b_j/a_j$. With $b_j \approx \bar{b}$, $\rho_j \propto 1/a_j$, hence small-gradient logits should be chosen first. Details are in Appendix A.

### 4.3 AUTO-OPTIMIZATION FRAMEWORK

To fully explore the search space introduced by SINAI, we formulate hyperparameter selection as a constrained multi-objective optimization problem. Section 4.3.1 specifies the decision variables along with their search space. Section 4.3.2 then develops a Bayesian optimization framework for constrained objective maximization, followed by implementation details.

#### 4.3.1 DECISION VARIABLES AND SEARCH SPACE

We define the decision variables per module type to reflect heterogeneous noise tolerance. For Softmax-based attention modules, we only search for the noise variance: $\theta_{\text{softmax}} \triangleq (\sigma_{\text{softmax}}), \sigma_{\text{softmax}} \geq 0$ over a broadened range, given their empirically higher robustness to injected noise. For ReLU/GeLU-based modules, we jointly tune the variance and the injection ratio: $\theta_{\text{act}} \triangleq (\sigma_{\text{act}}, \rho_{\text{act}}), \sigma_{\text{act}} \geq 0, \ \rho_{\text{act}} \in [0, 1]$ where $\rho_{\text{act}}$ controls the dynamic noise injection ratio for each logit. Thus, the overall decision vector: $\theta = (\sigma_{\text{softmax}}, \sigma_{\text{gelu}, \rho_{\text{gelu}}}, \sigma_{\text{relu}}, \rho_{\text{relu}})$ This formulation results in a larger and more complex search space than in prior work. To efficiently explore this space under the clean-accuracy constraint, we employ Bayesian optimization, described next in Section 4.3.2.

#### 4.3.2 BAYESIAN OPTIMIZATION AND IMPLEMENTATION

Given the decision variables above, we formulate hyperparameter tuning as a constrained multi-objective optimization problem, jointly optimizing clean and robust accuracy. During tuning process, we use both a clean validation dataset and an adversarial validation set generated by the *Square* attack (Andriushchenko et al., 2020), with a constraint that clean accuracy remains within a tolerance $\delta \in \{0.01, 0.02\}$ of the baseline model to prevent excessive degradation. Formally, for baseline clean accuracy $\text{Acc}_{\text{clean}}^{\text{base}}$, a candidate configuration $\theta$ is $\text{Acc}_{\text{clean}}(\theta) \geq \text{Acc}_{\text{clean}}^{\text{base}} - \delta$. Optimization is then performed over the feasible region by maximizing $(\text{Acc}_{\text{clean}}(\theta), \ \text{Acc}_{\text{rob}}(\theta))$, yielding a Pareto front that characterizes the trade-off between clean and robust performance.

We adopt Bayesian optimization with NSGA-II (Deb et al., 2002) (via Optuna's `NSGAIISampler` (Akiba et al., 2019)) to efficiently search the heterogeneous, activation function dependent space. Constraints are enforced by a feasibility-first strategy, where candidates violating the clean-accuracy bound are ranked by the degree of violation. To reduce cost, we employ multi-fidelity evaluation: early trials use reduced adversarial budgets, while promising ones are re-evaluated under the full protocol.

## 5 EXPERIMENTS

In this section, we present a comprehensive evaluation of our proposed method. We first describe the experimental setup (Section 5.1), followed by results against query-based black-box attacks (Section 5.2) and adaptive attacks (Section 5.3).

### 5.1 EXPERIMENTAL SETUP

**Datasets and Models.** We perform our experiments on two widely used benchmark datasets in adversarial robustness: CIFAR10 (Krizhevsky et al., 2009) and ImageNet (Russakovsky et al., 2015). Following prior work (Qin et al., 2021; Hung-Quang et al., 2024), we randomly select 1000 images that contain every class from the studied dataset in each experiment. For models, we consider two transformer-based architectures, including ViT (Dosovitskiy et al., 2020), DeiT (Touvron et al., 2021), as well as one CNN model, ResNet-50 (He et al., 2016). All models are loaded from the `timm` library, and we fine-tune them on CIFAR-10 using images resized to $224 \times 224$ resolution.

**Attacks.** For our main results, we follow prior work (Qin et al., 2021; Hung-Quang et al., 2024) and evaluate three representative score-based black-box attacks. Specifically, we consider two random-search methods, Square (Andriushchenko et al., 2020) and SignHunter (Al-Dujaili & O'Reilly,

Table 1: Performance on **CIFAR-10** (left) and **ImageNet** (right). Columns report clean accuracy (Clean) and robust accuracy under three score-based black-box attacks at two query budgets (1k/10k). Best values under each attack are in **bold**.

| | | CIFAR-10 | | | | | | | ImageNet | | | | | | |
| | | Clean | Square | | SignHunt | | NES | | Clean | Square | | SignHunt | | NES | |
| Model | Method | | 1k | 10k | 1k | 10k | 1k | 10k | | 1k | 10k | 1k | 10k | 1k | 10k |
|---|---|---|---|---|---|---|---|---|---|---|---|---|---|---|---|
| ResNet-50 | Base | 95.7 | 4.3 | 0.1 | 2.7 | 0.2 | 66.2 | 5.1 | 80.5 | 7.3 | 0.2 | 8.1 | 0.5 | 40.0 | 5.2 |
| | RND (≈1%) | 94.7 | 75.0 | 46.8 | 33.1 | 31.0 | **92.0** | 75.3 | 79.0 | 47.8 | 29.8 | 36.9 | 34.5 | 51.3 | 10.0 |
| | FND (≈1%) | 94.9 | 76.3 | 51.4 | 36.1 | 34.5 | 91.0 | 71.4 | 79.7 | 50.7 | 39.7 | 45.8 | 44.6 | 57.5 | 18.3 |
| | SINAI (≈1%) | 94.7 | **78.3** | **62.9** | **44.4** | **42.3** | 90.9 | **79.5** | 79.6 | **53.0** | **45.8** | **53.8** | **50.9** | **65.3** | **32.2** |
| | RND (≈2%) | 93.9 | 75.7 | 51.8 | 35.5 | 35.0 | 90.9 | 78.1 | 78.3 | 54.3 | 41.7 | 47.2 | 46.4 | 64.5 | 26.8 |
| | FND (≈2%) | 93.8 | 76.9 | 56.9 | 38.4 | 38.3 | 90.4 | 75.8 | 78.5 | 53.0 | 45.9 | 52.7 | 50.6 | 63.5 | 32.5 |
| | SINAI (≈2%) | 93.8 | **79.6** | **65.1** | **49.2** | **47.2** | **91.2** | **79.7** | 78.4 | **54.9** | **49.0** | **56.6** | **55.2** | **67.8** | **44.5** |
| ViT-B-16-224 | Base | 95.5 | 0.0 | 0.1 | 2.3 | 0.1 | 80.4 | 20.5 | 85.6 | 1.3 | 0.1 | 7.8 | 0.1 | 51.9 | 0.3 |
| | RND (≈1%) | 94.4 | 77.3 | 53.7 | **38.3** | 35.4 | 92.6 | 84.9 | 84.5 | 38.8 | 13.7 | 28.0 | 18.7 | 72.4 | 10.0 |
| | FND (≈1%) | 94.3 | 77.4 | 54.9 | 36.2 | 35.2 | 92.8 | 85.1 | 84.4 | 53.9 | 40.7 | 40.2 | 35.0 | **79.4** | 44.6 |
| | SINAI-Attn (≈1%) | 94.5 | **79.3** | **62.6** | 36.7 | **35.8** | **94.0** | **88.0** | 84.5 | **58.3** | **48.9** | **49.0** | **45.6** | 77.4 | 47.4 |
| | SINAI-FFN (≈1%) | 94.4 | 77.0 | 57.5 | 37.0 | 35.0 | 93.3 | 86.9 | 84.5 | 55.6 | 40.9 | 43.7 | 37.0 | 75.0 | 45.5 |
| | RND (≈2%) | 93.3 | 76.5 | 56.8 | 37.8 | 37.5 | 91.8 | 85.5 | 83.3 | 51.2 | 33.6 | 33.2 | 26.8 | 79.6 | 47.6 |
| | FND (≈2%) | 93.4 | 77.8 | 57.8 | 40.9 | 38.9 | 91.9 | 86.9 | 83.4 | 54.7 | 47.4 | 43.1 | 38.3 | **79.9** | 60.3 |
| | SINAI-Attn (≈2%) | 93.5 | **79.5** | **64.8** | **44.1** | **42.3** | **92.9** | 88.5 | 83.5 | **56.3** | **51.4** | **52.4** | **46.6** | 79.3 | **60.7** |
| | SINAI-FFN (≈2%) | 93.5 | 76.9 | 57.9 | 38.5 | 37.8 | 92.7 | **89.3** | 83.6 | 53.7 | 43.9 | 46.6 | 38.5 | 79.1 | 57.1 |
| DeiT-B-16-224 | Base | 97.7 | 0.9 | 0.1 | 8.8 | 0.0 | 73.1 | 7.4 | 81.4 | 0.0 | 0.1 | 1.7 | 0.2 | 48.9 | 0.0 |
| | RND (≈1%) | 96.9 | **87.6** | 69.6 | 57.8 | 56.5 | 95.3 | **87.9** | 80.3 | 45.2 | 30.7 | 27.9 | 22.6 | 76.9 | 52.3 |
| | FND (≈1%) | 96.8 | 86.2 | 71.5 | 58.1 | 56.5 | 94.4 | 86.2 | 80.3 | 51.6 | 38.5 | 33.0 | 30.1 | 75.3 | 53.7 |
| | SINAI-Attn (≈1%) | 96.9 | 86.7 | **75.9** | **61.9** | **59.3** | **96.8** | 87.2 | 80.2 | **52.0** | **45.3** | **36.1** | **34.7** | **78.8** | 59.4 |
| | SINAI-FFN (≈1%) | 96.8 | 86.0 | 74.7 | 59.1 | 57.4 | 96.4 | 87.1 | 80.5 | 51.3 | 40.9 | 33.7 | 30.9 | 78.7 | **61.0** |
| | RND (≈2%) | 95.4 | **88.6** | 75.9 | 60.8 | 59.2 | 94.6 | **89.9** | 79.4 | 46.8 | 35.3 | 29.2 | 24.9 | 77.0 | 54.0 |
| | FND (≈2%) | 95.6 | 83.8 | 75.4 | 56.0 | 54.8 | 93.7 | 88.1 | 79.3 | **51.2** | 43.2 | **39.2** | 33.4 | 75.7 | 55.6 |
| | SINAI-Attn (≈2%) | 95.6 | 85.3 | **76.5** | **60.9** | **59.6** | 95.3 | 88.5 | 79.5 | 50.2 | **44.9** | 37.4 | **35.2** | 78.0 | 62.3 |
| | SINAI-FFN (≈2%) | 95.6 | 85.2 | 73.6 | 60.5 | 58.2 | **95.5** | 86.9 | 79.6 | **51.2** | 40.6 | 36.1 | 34.2 | **78.6** | **63.4** |

2020), as well as a gradient-estimation method, NES (Ilyas et al., 2018a). In addition, we include two decision-based attacks, RayS (Chen & Gu, 2020) and Sign-Flip (Chen et al., 2020b).

**Defense Baselines.** We compare our method against two representative noise injection based defenses: input-level noise injection (RND) (Qin et al., 2021; Byun et al., 2022) and feature-level noise injection (FND) (Hung-Quang et al., 2024).

**Evaluation Protocol.** We report robust accuracy to measure the effectiveness of defenses. Consistent with FND, we constrain the drop in clean accuracy within 1% or 2% and search for the configuration that achieves the highest robust accuracy.

More details can be found in Appendix B.

## 5.2 EVALUATION AGAINST QUERY-BASED BLACK-BOX ATTACK

**Score-based Black-Box Attack.** Table 1 reports robustness under score-based black-box attack, where adversaries can obtain the model's output scores to guide the perturbations. These results confirm strategy-level conclusions aligned with our motivation. **(1)** Different activation functions require different noise strengths: injecting before Softmax (in attention) with stronger noise consistently yields a better clean–robust trade-off than before GeLU (in FFN). For example, when evaluated on two strong attack methods, Square and SignHunt, SINAI-Attn improves over FND by up to 7.0% on ViT-B-16-224 (CIFAR-10, Square-10k) and 8.3% on ImageNet under SignHunter-10k, while SINAI-FFN shows only marginal gains. **(2)** It's necessary to consider logit-level noise allocation in terms of ReLU/GeLU. Except for SINAI-FFN (GeLU), on ResNet-50 (CIFAR-10), SINAI achieves 65.1% robustness on Square-10k, while FND has only 56.9% under ≈2% clean accuracy drop. **(3)** Between ReLU and GeLU, ReLU layers benefit more from selective injection: on ResNet-50, SINAI improves ImageNet under SignHunt-10k robust accuracy to 55.2%, outperforming FND (50.6%) and RND (46.4%) under 2% clean accuracy drop. Overall, these findings demonstrate that module-aware, logit-level selective injection is useful for achieving a better trade-off in score-based black-box settings.

**Decision-based Black-Box Attack.** We further evaluate our method under the decision-based black-box attack methods. We evaluate ResNet-50 and ViT-B-16-224 and set our method and baselines as clean acc drop to ≈ 2%. As shown in Table 2, our method consistently outperforms baselines under RayS (Chen & Gu, 2020), benefiting from its fine-grained noise strategy rather than uniform one.

Table 2: ResNet-50 and ViT-B-16-224 performance on ImageNet under decision-based attacks.

| Model | Method | Clean Acc | RayS | SignFlip |
|-------|--------|-----------|------|----------|
| ResNet-50 | Base | 80.5 | 30.2 | 51.1 |
| | RND | 78.3 | 39.2 | **74.7** |
| | FND | 78.5 | 39.7 | 72.9 |
| | SINAI | 78.4 | **41.8** | 72.4 |
| ViT-B-16-224 | Base | 85.6 | 24.0 | 21.0 |
| | RND | 83.3 | 31.5 | **63.9** |
| | FND | 83.4 | 33.8 | 59.9 |
| | SINAI-Attn | 83.5 | **34.7** | 60.3 |
| | SINAI-FFN | 83.6 | 32.1 | 59.7 |

Table 3: ViT-B-16-224 performance on CIFAR-10 with various attack perturbations.

| Attack | $\mu$ | Small $\nu$ | | | | Large $\nu$ | | | |
|--------|-------|-----|-----|-----------|-----------|-----|-----|-----------|-----------|
| | | RND | FND | SINAI-Attn | SINAI-FFN | RND | FND | SINAI-Attn | SINAI-FFN |
| Square | 0.05 | 77.3 | 77.4 | **79.3** | 77.0 | 76.5 | 77.8 | **79.5** | 76.9 |
| | 0.10 | 39.6 | 42.5 | **52.3** | 48.9 | 43.7 | 45.8 | **51.4** | 45.4 |
| | 0.20 | 4.3 | 6.9 | **12.0** | 8.4 | 5.3 | 8.8 | **12.2** | 10.3 |
| SignHunt | 0.05 | **38.3** | 36.2 | 36.7 | 37.0 | 37.8 | 40.9 | **44.1** | 38.5 |
| | 0.075 | 18.3 | 18.4 | **26.4** | 22.9 | 21.9 | 20.2 | **27.9** | 22.4 |
| | 0.10 | 8.2 | 9.1 | **14.1** | 12.2 | 9.3 | 10.0 | **16.9** | 12.3 |
| NES | 0.05 | 92.6 | 91.9 | 92.9 | **93.3** | 91.8 | 86.9 | **94.0** | 92.7 |
| | 0.10 | 92.7 | 92.8 | **93.0** | 92.6 | 91.7 | 91.6 | **92.2** | 91.6 |
| | 0.20 | 92.8 | **92.9** | 92.8 | 92.7 | 91.4 | 91.6 | **92.2** | 91.5 |

In contrast, under SignFlip (Chen et al., 2020b), feature-level noise injection is less effective than input-level methods, which highlights that the success of noise injection defense is attack-dependent.

**Robustness under various perturbations.** To evaluate the scalability and stable performance of our method, we analyze the robustness of ViT-B-16-224 on CIFAR-10 under increasing perturbation magnitudes $\mu$ (Table 3). All baselines follow the same setting as before, allowing approximately $1\%$ or $2\%$ clean accuracy drop. As $\mu$ increases, the adversaries become stronger, leading to progressively larger degradation in robustness. While all defenses degrade, SINAI shows better scalability: SINAI-Attn consistently preserves the highest robustness under Square and SignHunt, particularly at larger $\mu$, whereas SINAI-FFN provides more stable performance. These results indicate that our strategy scales stably with attack strength than uniform-noise defenses, confirming its robustness in dynamic adversarial scenarios.

## 5.3 EVALUATION AGAINST ADAPTIVE ATTACK

Under adaptive attacks with Expectation Over Transformation (EOT) (Athalye et al., 2018), the attacker queries each sample $M$ times and averages the outputs to suppress randomness. We consider $M = 1, 5, 10$ and report results for both 1,000 queries and $M \times 1,000$ queries. Table 4 shows the robust accuracy of ViT-B-16-224 on ImageNet under Square, SignHunt, and NES attacks. The results reveal three key insights. First, robustness naturally decreases as $M$ grows, since averaging enables

Table 4: ViT-B-16-224 on ImageNet under three EOT score-based black-box attacks.

| Attacks | Methods | M=1 | $M = 5$ | | $M = 10$ | |
|---------|---------|-----|---------|---------|----------|----------|
| | | QC=1000 | QC=1000 | QC=5000 | QC=1000 | QC=10000 |
| Square | RND | 51.2 | 53.1 | 40.8 | 54.6 | 36.6 |
| | FND | 54.7 | 54.3 | 49.3 | 56.4 | 47.9 |
| | SINAI-Attn | **56.3** | **57.8** | **53.7** | **57.3** | **51.4** |
| | SINAI-FFN | 53.7 | 52.2 | 45.8 | 52.5 | 44.5 |
| SignHunt | RND | 33.2 | 29.5 | 16.5 | 34.4 | 11.1 |
| | FND | 43.1 | 36.7 | 25.6 | 39.3 | 21.1 |
| | SINAI-Attn | **52.4** | **47.0** | **40.7** | **48.1** | **32.4** |
| | SINAI-FFN | 46.6 | 42.4 | 32.1 | 41.6 | 28.6 |
| NES | RND | 79.6 | 82.0 | 76.9 | 82.9 | 75.4 |
| | FND | **79.9** | 81.5 | **78.6** | 82.3 | **77.7** |
| | SINAI-Attn | 79.3 | 82.0 | 75.7 | 83.1 | 72.8 |
| | SINAI-FFN | 79.1 | **82.4** | 75.0 | **83.5** | 72.8 |

the attacker to cancel injected noise more effectively. Second, SINAI-Attn consistently outperforms RND and FND across Square and SignHunt, highlighting the necessity of module-level noise consideration. Third, with our logit-level noise injection, SINAI-FFN performs comparably to FND under mild settings but loses advantage as $M$ increases, suggesting that perturbations in FFN layers are easier for the attacker to average.

## 6 CONCLUSION

In this paper, we introduced SINAI, a strategic noise injection method that achieves a better clean-robust trade-off under query-based black-box attacks. By studying module-level heterogeneity and logit-level gradient variations, SINAI allocates stronger noise to attention modules with Softmax and fine-grained selective injection to GeLU/ReLU. We further formulate the noise configuration search as a constrained optimization problem, which is solved automatically via Bayesian optimization. We expect SINAI to offer useful insights into how noise can be more effectively leveraged for model robustness. For future work, SINAI can be extended to other architectures, combined with training-based defenses, and evaluated under broader adversarial scenarios.

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

## A  PROOFS AND DETAILED DERIVATIONS

This appendix provides the full derivations and proofs that support Section 4.2.2.

**Robustness improvement coefficients.**  Following Theorem 1 in FND (Hung-Quang et al., 2024), the wrong-update probability of the attacker increases monotonically with the weighted sum of injected variances. Absorbing constants into per-logit coefficients, robustness is summarized as

$$\mathcal{R}(\sigma^2) = \sum_{j=1}^{m} b_j \sigma_j^2,$$

where

$$b_j \triangleq \mathbb{E}\big[(\partial_{z_j}(L \circ g))^2\big],$$

captures the marginal robustness gain from injecting noise into logit $j$.

**Clean degradation coefficients.**  Equation 2 expresses clean accuracy as a function of the margin rescaled by the variance of the noise. Expanding for small noise variances $\sigma_j^2$, a first-order Taylor approximation around $\sigma^2 = 0$ gives

$$\mathrm{Acc}_{\mathrm{clean}}(\sigma^2) = 1 - \sum_{j=1}^{m} a_j \sigma_j^2 + o(\|\sigma^2\|),$$

where each $a_j$ is the sensitivity of clean accuracy to the variance in logit $j$. Formally,

$$a_j \triangleq -\left.\frac{\partial}{\partial \sigma_j^2} \mathrm{Acc}_{\mathrm{clean}}(\sigma^2)\right|_{\sigma^2=0},$$

which scales with $(\partial_{z_j}(L \circ g))^2$ as stated in Equation 3. Intuitively, large gradients imply high sensitivity to injected noise and thus larger $a_j$.

**Optimization problem and Karush–Kuhn–Tucker(KKT) conditions.**  Combining the above, the allocation problem in Equation equation 4 is

$$\max_{\sigma_j^2 \geq 0} \sum_{j=1}^{m} b_j \sigma_j^2 \quad \text{s.t.} \quad \sum_{j=1}^{m} a_j \sigma_j^2 \leq C_0.$$

The Lagrangian is

$$\mathcal{L}(\sigma^2, \lambda) = \sum_{j=1}^{m} b_j \sigma_j^2 - \lambda\Big(\sum_{j=1}^{m} a_j \sigma_j^2 - C_0\Big).$$

KKT stationarity yields

$$\sigma_j^2 > 0 \;\Rightarrow\; b_j = \lambda a_j, \qquad \sigma_j^2 = 0 \;\Rightarrow\; b_j \leq \lambda a_j.$$

Hence the optimal allocation selects coordinates according to the ratio

$$\rho_j = \frac{b_j}{a_j}.$$

**Theorem A.1** (Small gradient first). *If $b_j$ varies slowly across logits, e.g. $b_j = \bar{b}(1 + \epsilon_j)$ with $|\epsilon_j| \leq \epsilon \ll 1$, then*

$$\rho_j = \frac{b_j}{a_j} \approx \frac{\bar{b}}{a_j}.$$

*Thus ordering by $\rho_j$ is equivalent to ranking by $1/a_j$, i.e. by selecting logits with the smallest $a_j$. Since $a_j \propto (\partial_{z_j}(L \circ g))^2$, the optimal allocation is to inject noise into logits with the smallest gradients; the rule is exact when $b_j$ is constant and $(1 - O(\epsilon))$-optimal otherwise.*

**Corollary A.2** (Better than uniform allocation). *Let $\mathcal{R}^\star(C_0)$ denote the optimal robustness under budget $C_0$, and $\mathcal{R}^{\mathrm{uni}}(C_0)$ the robustness from uniform allocation $\sigma_j^2 = \alpha$ with $\alpha = C_0/\sum_k a_k$. Then*

$$\mathcal{R}^\star(C_0) \geq \mathcal{R}^{\mathrm{uni}}(C_0),$$

*with equality iff all ratios $\rho_j = b_j/a_j$ are identical. Thus selective allocation strictly improves robustness over uniform noise injection in all heterogeneous cases.*

## B    Detailed Experimental Setup

**Attack Configurations.** For the attack configurations, we adopt different perturbation budgets depending on the dataset and model. On CIFAR-10, all attacks are constrained within an $\ell_\infty$ ball of radius $0.05$. On ImageNet, we use a radius of $0.05$ for ResNet-50 and a larger radius of $0.2$ for ViT-B and DeiT-B, reflecting the higher input resolution and robustness characteristics of transformer-based models.

**Optuna Hyperparameter Search.** We employ Optuna with the NSGA-II sampler (Akiba et al., 2019) to conduct multi-objective hyperparameter search. The objectives are to maximize both clean accuracy and robust accuracy simultaneously. Specifically, we tune the noise variance parameter $\sigma$ within different ranges for different modules using 40 trials. Based on the clean accuracy trends in our motivation, we set the search space of noise variance $\nu$ according to the point where accuracy begins to degrade significantly. For ViT-B-16-224, the ranges are $\nu \in [0, 0.1]$ for RND, $\nu \in [0, 0.1]$ for FND, $\nu \in [0, 2.0]$ for attention module, and $\nu \in [0, 0.4]$ for FFN module injection. For ResNet-50, the ranges are $\nu \in [0, 0.1]$ for RND, $\nu \in [0, 0.1]$ for FND, and $\nu \in [0, 0.4]$ for pre-ReLU injection. In addition, for ReLU and GeLU modules we tune the noise injection ratio within $[0, 100]\%$, ensuring that the search jointly explores both the variance and the proportion of perturbed logits. Each trial constructs a robust model with the candidate noise setting and evaluates two metrics: (1) clean accuracy on the evaluation set, averaged over $M = 3$ Monte Carlo runs, and (2) robust accuracy under the Square attack with a budget of 1000 queries. A similar process is conducted for DeiT-B.

**Defense Configurations.** We employ Optuna with NSGA-II to jointly search the noise variance $\nu$ and injection ratio, selecting configurations that achieve either $\approx1\%$ or $\approx2\%$ clean accuracy drop. The search ranges are guided by our motivation experiments (Figures in Section 5.1), where the upper bound is chosen near the turning point of clean accuracy degradation. The final configurations selected on CIFAR-10 and ImageNet are summarized in Tables 5 and 6, respectively. Notably, Optuna consistently assigns higher $\nu$ values to Softmax than to GeLU or ReLU modules, confirming that attention components are intrinsically more tolerant to injected noise.

Table 5: Hyperparameter configurations selected by Optuna on **CIFAR-10**. Values correspond to noise variance $\nu$ and injection ratio (for SINAI). Each configuration achieves either $\approx1\%$ or $\approx2\%$ clean accuracy drop.

| Model | RND | | FND | | SINAI | |
|---|---|---|---|---|---|---|
| | $\nu$ (1%) | $\nu$ (2%) | $\nu$ (1%) | $\nu$ (2%) | $\nu$/ratio (1%) | $\nu$/ratio (2%) |
| ResNet-50 | 0.05 | 0.06 | 0.14 | 0.17 | ReLU: 0.20 / 52% | ReLU: 0.23 / 67% |
| ViT-B-16-224 | 0.08 | 0.10 | 0.09 | 0.11 | Softmax: 1.00; GeLU: 0.14 / 68% | Softmax: 1.20; GeLU: 0.16 / 67% |
| DeiT-B-16-224 | 0.07 | 0.10 | 0.22 | 0.28 | Softmax: 1.20; GeLU: 0.28 / 60% | Softmax: 1.44; GeLU: 0.31 / 75% |

Table 6: Hyperparameter configurations selected by Optuna on **ImageNet**. Values correspond to noise variance $\nu$ and injection ratio (for SINAI). Each configuration achieves either $\approx1\%$ or $\approx2\%$ clean accuracy drop.

| Model | RND | | FND | | SINAI | |
|---|---|---|---|---|---|---|
| | $\nu$ (1%) | $\nu$ (2%) | $\nu$ (1%) | $\nu$ (2%) | $\nu$/ratio (1%) | $\nu$/ratio (2%) |
| ResNet-50 | 0.01 | 0.02 | 0.10 | 0.15 | ReLU: 0.23 / 37% | ReLU: 0.22 / 72% |
| ViT-B-16-224 | 0.02 | 0.05 | 0.04 | 0.06 | Softmax: 0.70; GeLU: 0.06 / 70% | Softmax: 1.00; GeLU: 0.11 / 74% |
| DeiT-B-16-224 | 0.05 | 0.07 | 0.13 | 0.20 | Softmax: 1.17; GeLU: 0.15 / 68% | Softmax: 1.28; GeLU: 0.23 / 83% |

## C    Ablation Study

**Effect of Noise Variance.** To further understand the role of noise magnitude, we conduct experiments on ImageNet by varying the noise variance $\nu$ for different modules. Unlike the motivation experiments in Section 5.1, which focused only on clean accuracy, here we also track robust accuracy under black-box attacks. This allows us to characterize the trade-off between clean and robust performance as $\nu$ increases, and to verify that modules differ in their tolerance to noise injection. The results confirm our motivation: modules with different activations exhibit distinct robust-

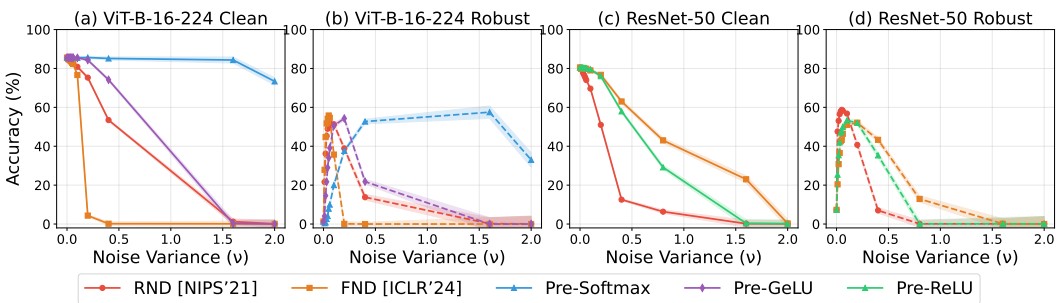

Figure 3: (a)(b) ViT-B-16-224 clean and robust accuracy under various noise variances with noise injection to different activation functions. (c)(d) ResNet-50 clean and robust accuracy under various noise variances.

ness–accuracy trade-offs. For ViT-B-16-224, injecting before Softmax shows stronger robustness compared to GeLU, validating that attention modules are more noise-tolerant (Figure 3(a)–(b)). By contrast, GeLU layers in FFNs collapse rapidly as $\nu$ increases, reflecting their higher noise sensitivity. For ResNet-50, injecting before ReLU maintains higher noise sensitivity (Figure 3 (c)). These findings show our first conclusion that noise magnitude must be module-aware.

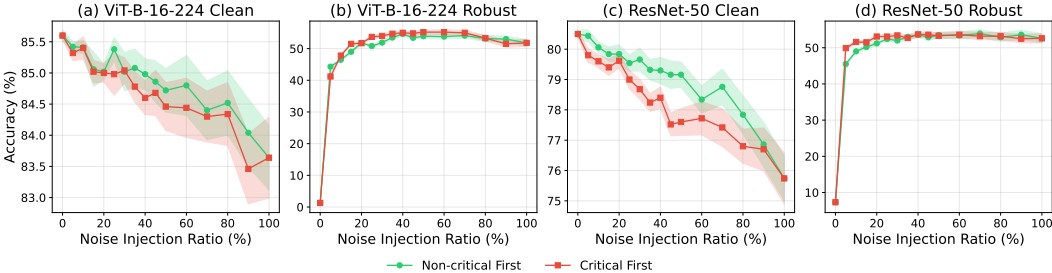

Figure 4: (a)(b) ViT-B-16-224 clean and robust accuracy under various noise injection ratios with noise injection to GeLU. (c)(d) ResNet-50 clean and robust accuracy under various noise injection ratios.

**Effect of Injection Ratio.** We additionally study the injection ratio for ReLU and GeLU functions. By varying the ratio from $0\%$ to $100\%$, we examine how selectively injecting noise impacts both clean and robust accuracy. This analysis provides finer-grained insight into how GeLU and ReLU modules respond when noise is applied to different fractions of logits. As shown in Figure 4 (a),(b), for ViT-B-16-224, starting from non-critical logits preserves clean accuracy while achieving similar robustness as critical-first allocation, consistent with the observation that most GeLU logits are small. For ResNet-50, the difference is more pronounced: critical-first injection causes a sharper drop in clean accuracy, whereas non-critical-first maintains accuracy better while still improving robustness (Figure 4 (c) and (d)). These results confirm that selective allocation is essential and that ReLU layers gain larger advantages than GeLU when noise is injected in a logit-aware manner.

## D    USE OF LLMS

We only use LLMs for sentence-level editing and proofreading of the manuscript.

## E    LIMITATIONS AND FUTURE WORK

Our study has two main limitations. First, we primarily evaluate noise injection on image classification with ViTs and CNNs, leaving its generalization to multimodal large models or non-classification tasks such as detection and segmentation unclear. Second, our evaluation is restricted to query-based

black-box attacks, and it remains open how well the method extends to other black-box scenarios, such as transfer-based attacks.

Future work can extend our method beyond vision models to multimodal and language models, since our analysis provides direct insights into how noise may be applied to architectures with similar activation functions. Another important direction is to investigate the role of noise across broader adversarial scenarios and downstream tasks, in order to better understand its generality and practical limitations.

