# OpenReview forum: "SINAI: Strategic Injection of Noise for Adversarial  Defense with Improved Accuracy–Robustness Tradeoffs"
_ICLR.cc/2026/Conference — Submitted to ICLR 2026_

### Official Review · Reviewer_15EF · 2025-10-29

**Soundness:** 3
**Presentation:** 3
**Contribution:** 3
**Rating:** 6
**Confidence:** 2

**Summary:**

The paper proposes SINAI, a defense mechanism against query-based black-box adversarial attacks that strategically injects noise during inference. By revealing that different network modules exhibit heterogeneous noise sensitivities depending on their activation functions, and that gradient norms vary across logits, the method introduces a logit-aware, module-specific noise allocation strategy. Furthermore, SINAI employs a Bayesian optimization framework to automatically search for noise magnitudes under a clean-accuracy constraint, achieving consistent improvements on CIFAR-10 and ImageNet compared with existing input- and feature-level noise injection baselines.

**Strengths:**

1. The paper provides a well-structured motivation, connecting empirical sensitivity analysis to theoretical modeling.

2. The combination of module-aware and logit-level injection is novel and technically grounded.

3. The optimization framework using NSGA-II Bayesian search is interesting and improves reproducibility compared with purely manual tuning.

4. The experimental results of this article are very convincing.

**Weaknesses:**

1. All evaluations remain intra-dataset and intra-model. Therefore, this paper does not demonstrate transfer learning or cross-domain robustness.

2. The robustness improvement is mainly validated under query-based attacks.

3. Runtime and computational cost are not quantified.

**Questions:**

1. See in W1, how well does SINAI generalize to transfer-based attacks or cross-dataset scenarios?

2. Could the proposed logit-level selection be integrated into a learnable module or jointly optimized during adversarial training?

3. Have you tested robustness when using adversarially trained or robust pretrained backbones instead of clean ones?

4. Does SINAI introduce randomness that may reduce reproducibility between repeated inference runs?

5. What is the computational overhead of Bayesian optimization and the additional noise sampling during inference?

**Details Of Ethics Concerns:**

No.

---

### Official Review · Reviewer_xXLT · 2025-10-31

**Soundness:** 3
**Presentation:** 3
**Contribution:** 2
**Rating:** 4
**Confidence:** 3

**Summary:**

The paper proposes SINAI (Strategic Injection of Noise), a framework for enhancing robustness of deep models under black-box query-based attacks through selective and module-aware noise injection. Unlike prior methods (e.g., random noise injection [RND] or feature noise defense [FND]) that add uniform perturbations across the model, SINAI strategically injects noise:

At the module level — differentiating between Softmax-attention and activation functions (GeLU/ReLU) based on noise sensitivity.

At the logit level — injecting noise preferentially into “non-critical” (low-gradient) neurons.

The noise parameters (variance σ, ratio, etc.) are auto-tuned via constrained Bayesian optimization to maintain ≤2% clean accuracy drop.

Extensive experiments on CIFAR-10 and ImageNet with ViT/ResNet architectures show consistent improvements (≈4% robust accuracy gain under Square, SignHunter, and NES attacks) compared to FND and RND baselines.

**Strengths:**

1. Clear Motivation and Novelty:
The authors present a strong empirical and theoretical case for heterogeneous noise tolerance across modules. The distinction between pre-Softmax and pre-activation noise is both intuitive and underexplored in robustness research.

2. Comprehensive Evaluation:
The evaluation covers multiple black-box attack types (score-based, decision-based, and adaptive-EOT). The paper reports consistent robustness improvements across ViT, DeiT, and ResNet families.

3. Auto-tuning Framework:
The constrained Bayesian optimization for noise scheduling is practical and generalizable. It automatically balances clean vs. robust accuracy trade-offs without manual tuning.

4. Empirical Insightfulness:
The results and ablations (e.g., Figures 1–4) offer meaningful insight: Softmax is most noise-tolerant, and logit-level heterogeneity is substantial. These findings may influence other research beyond noise defenses.

**Weaknesses:**

1. Evaluation scope:
Experiments are limited to image classification. The claims of generality (e.g., “module-aware noise strategies apply to any architecture”) would be more convincing if supported by results on other modalities (e.g., diffusion models, NLP, or vision-language transformers).

2. Limited adaptive attack analysis:
The adaptive EOT evaluation uses relatively shallow ensembles (M ≤ 10).
Recent works on query-efficient adaptive attacks (e.g., Bandit, NES++, ZO-Attack) could better test SINAI’s robustness.

3. Optimization cost and practicality:
The paper briefly mentions multi-fidelity Bayesian optimization but does not quantify compute overhead. Practitioners may find it challenging to replicate SINAI’s tuning without clearer cost analysis or ablation on optimization budget.

4. Lack of comparison beyond noise defenses:
While SINAI’s relative gains over FND/RND are clear, it would strengthen the paper to compare against non-noise black-box defenses (e.g., ensemble smoothing, randomized smoothing, or confidence-based query filtering).

**Questions:**

1. How does SINAI behave under transfer-based black-box attacks, where query budget isn’t a bottleneck?

2. How sensitive is the performance to the number of noise-injected modules? Could partial (e.g., top-2 layers only) injection achieve similar results?

3. Did you explore per-layer adaptive σ learned directly via gradients (e.g., through a small meta-network)?

---

### Official Review · Reviewer_kwzG · 2025-11-01

**Soundness:** 2
**Presentation:** 2
**Contribution:** 2
**Rating:** 2
**Confidence:** 3

**Summary:**

In this paper, the authors propose the Strategic Injection of Noise for Adversarial Defense (SINAI) method, designed to achieve improved accuracy-robustness trade-offs. The approach accounts for the influence of activation functions during noise injection and employs a fine-grained, logit-aware allocation strategy. Furthermore, SINAI formulates the hyperparameter search for noise injection as a constrained optimization problem solved via Bayesian optimization. Experiments conducted on various CNN and ViT models using the CIFAR-10 and ImageNet benchmark datasets demonstrate that SINAI achieves superior robust accuracy compared to baseline defense methods.

**Strengths:**

1. This paper proposes a strategic noise injection method based on a fine-grained, logit-aware allocation strategy to improve the accuracy–robustness trade-off.

2. The paper formulates the hyperparameter search for noise injection as a constrained optimization problem and solves it using a Bayesian optimization algorithm.

3. Experimental results show that the proposed SINAI method outperforms existing approaches in terms of robust accuracy.

**Weaknesses:**

1. The equation presented in Section 4.2.1 lacks sufficient explanatory context. Regarding the variance approximation, the authors directly provide the approximate equation without explaining its derivation process or the underlying assumptions.

2. The equations presented in Section 4.2.2 are somewhat ambiguous and lack sufficient motivation and explanation. Specifically, the authors mention “absorbing constants into per-logit coefficients” but do not clarify the rationale behind this derivation of Equation (1). Moreover, compared with FND [1], Equation (1) appears to be a modification of the earlier formulation without demonstrating clear theoretical innovation.

3. In Theorem 1, the assumption that $b_j$ varies slowly across logits lacks sufficient justification. The paper presents this assumption without explaining its rationale or providing theoretical analysis or experimental evidence to support it.

4. The implementation details of the optimization algorithm in Section 4.3 are insufficiently described, making its logic difficult to follow. Moreover, the absence of complexity and convergence analyses hinders a proper assessment of the algorithm’s stability and robustness.

5. While the paper adopts Bayesian optimization with NSGA-II, it provides neither a rationale for this choice nor comparative experimental analysis. For example, it remains unclear how this algorithm ensures that the search process converges to an optimal solution. Moreover, the absence of further analysis leaves the effectiveness of this optimization approach unverified.

6. The experimental results are limited and lack comprehensiveness. Only three surrogate models are evaluated, which is insufficient to demonstrate the broad applicability of the proposed method. Moreover, as shown in Tables 1–4, the proposed SINAI method exhibits suboptimal performance under certain experimental configurations, whereas prior works (e.g. FND) achieve stronger defense results.

7. The experimental comparison should include a broader range of attack settings, as prior studies (e.g., RND) have incorporated comparisons with other representative approaches such as Sign-OPT [1], HSJA [2], and GeoDA [3]. In addition, recent black-box attack methods, such as SemiAdv [4] and P-BO [5], should also be considered for a more comprehensive evaluation.

[1] Sign-opt: A query efficient hard-label adversarial attack. ICLR 2020.

[2] HopSkipJumpAttack: A Query-Efficient Decision-Based Attack. SP 2020.

[3] Geoda: a geometric framework for black-box adversarial attacks. CVPR 2020.

[4] SemiAdv: Query-Efficient Black-Box Adversarial Attack with Unlabeled Images. ArXiv 2024.

[5] Efficient Black-box Adversarial Attacks via Bayesian Optimization Guided by a Function Prior. ICML 2024.

**Questions:**

1. Could the authors provide a more detailed explanation of the motivation and underlying assumptions discussed in Section 4.2.2?

2. How does the Bayesian optimization combined with NSGA-II search the optimal solution of optimization problem (4)? Additionally, what is the time complexity of this algorithm compared with other defense methods? How the $\rho$ control the noise injection intensity?

3. How does the proposed SINAI perform compared with state-of-the-art methods when evaluated on other CNN and ViT architectures?

4. How does the proposed SINAI perform under different evaluation settings against representative black-box attack methods, such as Sign-OPT, HSJA, GeoDA, SemiAdv, and P-BO?

---

### Official Review · Reviewer_MyGY · 2025-11-04

**Soundness:** 3
**Presentation:** 3
**Contribution:** 3
**Rating:** 6
**Confidence:** 3

**Summary:**

The paper introduces SINAI, a novel defense method for defending against query-based black-box attacks at inference time through strategic noise injection while maintaining clean accuracy. The paper puts especial emphasis on the characteristics of Vision Transformers (ViT), noticing that the heterogeneity between attention and feedforward modules deserve different treatment for the noise injection. The authors observe that different modules (e.g., and even individual logits within a layer respond differently to injected noise, largely due to the properties of their activation functions (Softmax vs GeLU/ReLU) and gradient norms. Then, relying on these observations, the authors propose SINAI to adaptively inject stronger noise into noise tolerant attention modules and finer, logit-aware noise into more sensitive FFN/CNN layers. SINAI prioritize less critical logits with small gradients to minimize the loss in accuracy. The parameters of the algorithm for the optimal noise configuration are searched automatically using a Bayesian optimization framework that balances clean and robust performance. Experiments on CIFAR-10 and ImageNet with different model architectures (including ViT, DeiT, and ResNet) show that SINAI outperforms other noise injection defenses, improving the robust accuracy while minimizing the clean accuracy drop against different query-based black box attacks.

**Strengths:**

+ The problem is well motivated and shows the limitations of previous noise injection approaches to defend against adversarial examples in black-box settings. The paper motivates well the necessity for non-uniform noise injection across modules and logits, especially for ViT architectures, where the differences between attention and feedforward modules are more prominent.
+ The derivation in equations (1)-(4) formalizes well the trade-off between robustness and clean accuracy. The “small-gradient-first” rule provides an intuitive justification for the selective noise injection and is well connected to the activation sensitivity.
+ The approach for SINAI algorithm is interesting: it combines a global, data-set level optimization (via Bayesian optimization and the use of a validation dataset) with an input-specific activation-based noise selection. This avoids per-input gradient computations while maintaining adaptivity to different types of inputs.
+ Overall, the experiments provide a comprehensive evaluation of query-based black-box attacks, including both CNNs and ViT architectures, two computer vision benchmarks (CIFAR-10 and ImageNet), and include adaptive attacks with EOT.
+ I appreciate the authors framed well the scope of the paper to query-based black-box attacks, and not as a universal adversarial defense. I think this clarity helps to contextualize better the results.

**Weaknesses:**

+ In Figure 2(b)/(c) in Section 3, some of the observed differences seem relatively small given the error bars and no statistical significance is provided to justify well the observations from these two plots. Thus, the claims about “substantial” differences between critical and non-critical logits are not well supported.
+ The paper misses the analysis of the computational overhead of the proposed method, considering the cost of Bayesian optimization vs the per-sample logit selection at inference time and observe what is the extra cost compared to other competing baselines.
+ No comparisons are made with other types of defenses non-relying on noise injection (e.g., adversarial training, randomized smoothing, etc.). In this sense, even a brief result against a strong adversarially trained model would help to contextualize better SINAI’s robustness.
+ Although the exclusive focus on query-based black box attacks is consistent with the paper’s defined goal, it limits the insights into SINAI’s behavior against other type of attacks, like white-box or transfer attacks. Even a short analysis would make the paper stronger.

**Questions:**

+ About the statistical significance of the results in Figure 2(b)/(c), it seems that the performance gap between noise injection on critical vs non-critical logits appears within the error bars. Have you run multiple random seeds or statistical tests to confirm the significance of the results?
+ Can you provide more details in the computational cost of the whole optimization process? How does this compare to other noise-injection methods in the related work?
+ Since critical logits are those with high sensitivity, could black-box attacks implicitly learn to target these regions more efficiently? In this sense, have you tested attack strategies that explicitly maximize the perturbation on the top-n critical logits?

---

### Meta-Review · Area_Chair_6mGE · 2026-01-05

**Summary:**

The reviewers raised several concerns, including the experiments being limited, lack of computional cost analysis, lack of compared methods, lack of sufficient justification of the proposed method, missing implementation details, etc. The authors did not provide an informative rebuttal, failing to address the core concerns raised.

**Reviewer Concerns:**

The authors did not provide an informative rebuttal, failing to address the core concerns raised.

**Reviewer Scores:**

The authors did not provide an informative rebuttal, failing to address the core concerns raised.

---

### Decision · Program_Chairs · 2026-01-26

Reject